# Real-Time Quantitative Ultrasound and Radar Knowledge-Based Medical Imaging

**Tom Sharon**                                                     TOM.SHARON@WEIZMANN.AC.I
**Hila Naaman**                                                   HILA.NAAMAN@WEIZMANN.AC.I
**Yonathan Eder**                                                    YONI.EDER@WEIZMANN.AC.I
**Yonina C. Eldar**                                               YONINA.ELDAR@WEIZMANN.AC.I
*Faculty of Math and Computer Science, Weizmann Institute of Science, Rehovot, Isreal*

## Abstract

Ultrasound and radar signals are useful for medical imaging due to their non-invasive, non-radiative, low-cost, and accessible nature. However, traditional imaging techniques lack resolution, contrast, and physical interpretation. Quantitative medical imaging is necessary for this purpose, as it provides a visual representation of physical characteristics. However, current techniques have drawbacks, including convergence to local minima and delayed results, which can lead to unsatisfactory outcomes. To address these limitations, we propose a neural network that incorporates the symmetries and properties of the received signals to achieve real-time quantitative mappings of physical properties. Our method achieves high accuracy using several numerical metrics for complex shapes with less than 0.15 seconds per test sample, compared to 0.75-2 hours for the competing method.

**Keywords:** ultrasound, radar, quantitative imaging, U-net, FWI, ISP, channel data

## 1. Introduction

Ultrasound (US) and radar signals are two primary signals used in non-invasive and non-radiative medical imaging. US and radar imaging produce images based on the received signals, referred to as Channel Data (CD), obtained from the received echoes (for US) or microwave signals (for radar) scattered by the scanned medium (Van Veen and Buckley, 1988; Li et al., 2005), as illustrated in Fig. 1.a-c. However, current imaging techniques using US or radar signals provide limited physical interpretation, hence Quantitative Medical Imaging (QMI) is needed. QMI has the potential to visualize various Quantitative Physical Properties (QPPs) of the scanned medium, such as Speed-of-Sound (SoS), density, and relative permittivity, which can be beneficial for diverse medical applications such as fatty liver diagnosis, and stroke imaging (Ruby et al., 2019; Ireland and Bialkowski, 2011).

To obtain quantitative images, a non-linear Inverse Scattering Problem (ISP) must be solved for reconstructing the QPPs of the scanned medium using the received CD. Full Waveform Inversion (FWI) is one such method that utilizes an iterative optimization algorithm to solve the ISP (Shultzman and Eldar, 2022; Guasch et al., 2020). The goal of FWI is to minimize the loss between the measured and predicted CD according to the QPPs estimation. However, FWI can be time-consuming and may converge to a local minimum. Furthermore, an initial guess close to the real solution based on some prior knowledge is required, which is often unavailable in realistic settings. To address these limitations, deep learning techniques have been proposed as a potential solution for solving the ISP (Chen

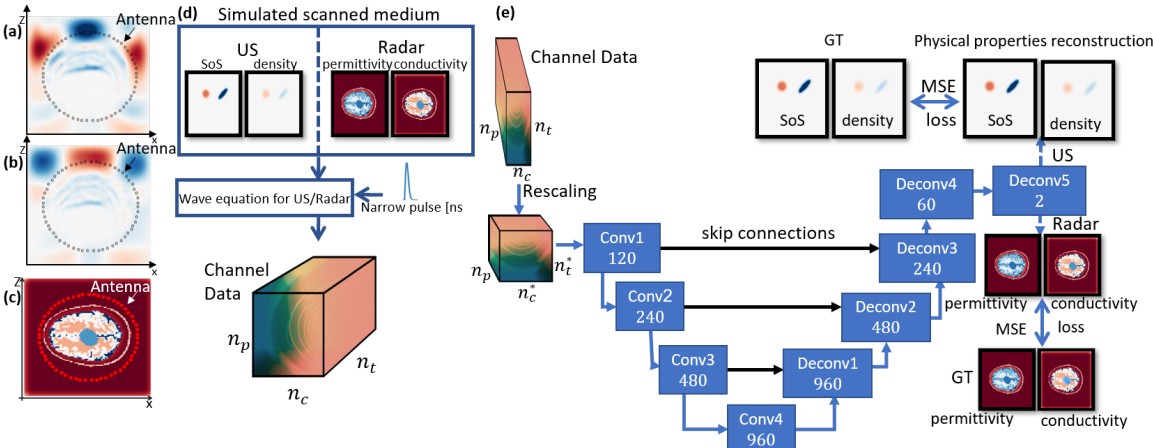

Figure 1: (a)-(b) depict the wave propagation from one antenna over two successive time samples. (c) depicts a medium with 60 antennas surrounding a stroke-affected brain. (d) displays the training set creation process from simulation, utilizing the QPPs of the grid and known pulse to obtain the channel data for network input. (e). QUARK-MI architecture for both radar and US schemes. The channel number of each convolution block is displayed.

et al., 2020; Wei and Chen, 2018; LeCun et al., 1998). Nevertheless, previous works only reconstructed one QPP and tested it on simple synthetic tests.

Our contribution is twofold. First, we propose a Neural Network (NN) for real-time reconstruction of multiple QPPs from measured CD. By incorporating the physical meaning of the input CD, our method achieves more accurate reconstructions and avoids converging to a local minimum, as occurs in FWI. Second, we demonstrate the versatility of our proposed network by showing its ability to reconstruct physical properties using either radar signals or US signals. Additionally, we demonstrate the network's ability to reconstruct more than one property for each case, and to handle complex nonhomogeneous domains such as a realistic human brain with random stroke.

## 2. Data and Method

We introduce QUARK-MI (Quantitative-Ultrasound-and-Radar Knowledge-based Medical Imaging), a real-time NN solution that can accurately reconstruct the medium's QPPs from either US or radar signals. QUARK-MI has a U-Net based architecture with skip connections, stride convolution, and batch normalization, enabling the NN to learn from different signal channels and capture fine details while preserving information from the entire CD tensor. The input to the NN is the CD tensor consists of the receiving signals over time for each transmission. Our training sets include simulations of one random object in a sample with liver properties for the US case, and different training sets simulations for the radar case: MNIST digit (LeCun et al., 1998) in a random position and blood properties as the scatter object, and realistic brain slices generated according to (Qureshi and Mustansar, 2017), with a random stroke. We use the known wave propagation equations to create the CD inputs (Fig. 1.d).

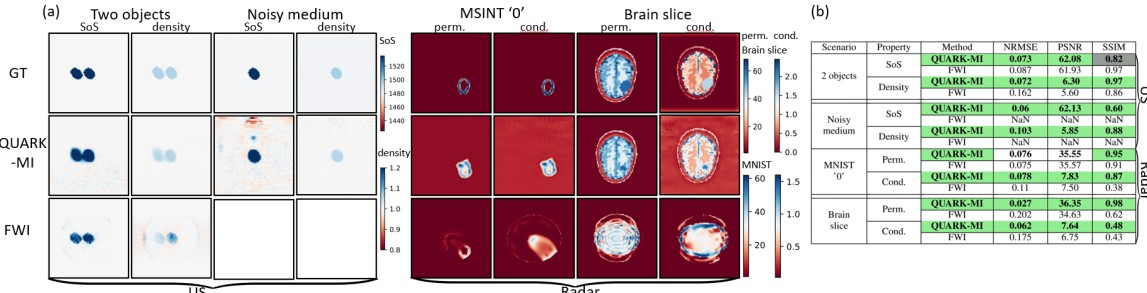

Figure 2: (a) The comparison of QUARK-MI and FWI methods for the reconstruction with respect to the Ground-Truth (GT). (b) The accuracy performance.

| Scenario | Property | Method | NRMSE | PSNR | SSIM |
|---|---|---|---|---|---|
| 2 objects | SoS | QUARK-MI | **0.073** | **62.08** | 0.82 |
| | | FWI | 0.087 | 61.93 | 0.97 |
| | Density | QUARK-MI | **0.072** | **6.30** | **0.97** |
| | | FWI | 0.162 | 5.60 | 0.86 |
| Noisy medium | SoS | QUARK-MI | **0.06** | **62.13** | **0.60** |
| | | FWI | NaN | NaN | NaN |
| | Density | QUARK-MI | **0.103** | **5.85** | **0.88** |
| | | FWI | NaN | NaN | NaN |
| MNIST '0' | Perm. | QUARK-MI | **0.076** | 35.55 | **0.95** |
| | | FWI | 0.075 | 35.57 | 0.91 |
| | Cond. | QUARK-MI | **0.078** | **7.83** | **0.87** |
| | | FWI | 0.11 | 7.50 | 0.38 |
| Brain slice | Perm. | QUARK-MI | **0.027** | **36.35** | **0.98** |
| | | FWI | 0.202 | 34.63 | 0.62 |
| | Cond. | QUARK-MI | **0.062** | **7.64** | **0.48** |
| | | FWI | 0.175 | 6.75 | 0.43 |

Our approach integrates the CD physical interpretation into the network architecture. Each transmission channel offers complementary information about the QPPs and treated as the convolution channels, while the time and receiving signal dimensions are treated as the spatial image dimensions in traditional U-Net. Additionally, as illustrated in Fig.1.e, we employ derivable rescaling to achieve square spatial dimensions. The final convolution block in our model sums the transmission channels over the spatial dimensions, generating two channels that depict two mappings of the medium's QPPs.

## 3. Evaluation, Results and Conclusion

**Evaluation and results**   We conducted a comprehensive evaluation of our QUARK-MI method and compared it with the FWI algorithm using numerical metrics (NRMSE, PSNR, and SSIM), to assess the accuracy of object shape and position as well as quantitative values of pixels, as shown in Fig.2. Our NN accurately reconstructs QPPs from challenging US signals scenarios, including two scattering objects with a uniform background (trained on only one object in each sample), and a noisy medium (where FWI diverge). Additionally, for the radar case, we reconstructed non-defined objects from the MNIST dataset with QPPs, and a realistic brain slice with a generated stroke. QUARK-MI outperforms FWI in most scenarios (highlighted in green), except for one case where FWI has as the initial guess the homogeneous background value (highlighted in grey). Our NN shows a generalization ability to reconstruct multiple objects even though the training set contained only one random object in each image, and reconstruction of highly complex shapes, even in the case of a human brain where the skull causes a significant decrease in signal quality, in which both the outline of the brain and the location and size of the stroke were restored with great success. Our method achieves high accuracy in complex and realistic objects, with less than 0.15 seconds per sample, compared to 0.75-2 hours for FWI.

**Conclusion**   In conclusion, our proposed NN that incorporates the symmetries and properties of received signals can achieve real-time mappings of QPPs for medical imaging. This method provides improved physical interpretation and can lead to the accomplishment of new clinical goals such as fast stroke imaging and cancer detection. With high accuracy and fast computational time, this approach has the potential to significantly impact the field of medical imaging.

## Acknowledgment

This research was supported by the European Research Council (ERC) under the European Union's Horizon 2020 research and innovation program (grant No. 101000967)and by the Israel Science Foundation (grant No. 3805/21) within the Israel Precision Medicine Partnership program.

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
