# OpenReview forum: "Real-Time Quantitative Ultrasound and Radar Knowledge-Based Medical Imaging"
_MIDL.io/2023/Short_Paper_Track — MIDL 2023 Short paper track Poster_

### Official Review · Reviewer_jBjD · 2023-04-14
**interesting work but very preliminary**

**Rating:** 6
**Confidence:** 5

**Review:**

This is an interesting short paper with great potential. There is no novelty in the developed CNN architecture. However, the application area is interesting with significant improvements. Important details are missing: For example I was very confused how the brain slices were obtained? Is this simulated data? Ultrasound signals cannot penetrate through the skull. What does MSINT '0' stand for?

---

### Official Review · Reviewer_2QPQ · 2023-04-24
**This paper proposes a deep learning-based neural network to fast reconstruct ultrasound (US) and radar images. More specifically, the authors incorporate the symmetries and properties of the received channel data (CD)  in the network design through convolutional layers in a traditional Unet.**

**Rating:** 9
**Confidence:** 3

**Review:**

Strength:

-The proposed work of fast ultrasound and radar image reconstruction with decent quality is interesting and could potentially have a high impact in the medical domain.

-The idea of treating CD data as convolutional channels seems novel.

-The paper is well written and is easy for readers to follow.

Weaknesses:

-While this is a short paper submission, the experimental validation could be further strengthened. There are many other advanced network architectures (i.e., transformers and diffusion networks) that could potentially achieve better performance than Unet. It would be interesting to see how much the proposed network is better than those works.

-It would be interesting to visualize the reconstructed examples of all methods (if the space allows).